Ecological features and insecticide resistance of Aedes albopictus in Xi’an, a high-risk dengue transmission area in China

Lei Xiaogang 1
Pang Songtao 1
Zhang Qipeng 2
Xu Kun 1
Xue Wei 1
Wu Mingxu 1
Li Xiangdong 1
Jin Liangdong 1
Li Guangshuai 1
Chen Baozhong 578356378@qq.com 1
1 Department of Vector Disease Control and Prevention, Xi’an Center for Disease Control and Prevention , Xi’an , Shaanxi , China
2 Department of Hospital Infection Management, The Affiliated Hospital of Zunyi Medical University , Zun Yi , Guizhou , China
Brygadyrenko Viktor
Electronic publication date: 2024 Oct 22
Publication date: 2024
Volume: 12
Electronic Location ID: e18246
Received 2024 Jul 8; Accepted 2024 Sep 15
Copyright: ©2024 Lei et al.
Copyright year: 2024
Copyright holder: Lei et al.
License: This is an open access article distributed under the terms of the Creative Commons Attribution License, which permits unrestricted use, distribution, reproduction and adaptation in any medium and for any purpose provided that it is properly attributed. For attribution, the original author(s), title, publication source (PeerJ) and either DOI or URL of the article must be cited.
License URL: https://creativecommons.org/licenses/by/4.0/

Keywords: Insecticides resistance, Aedes albopictus, Dengue, Meteorological factors

Funding: The authors received no funding for this work.

==============================
Background

Dengue, a mosquito-borne viral disease, has occurred in many cities in China, and it tends to spread to higher latitudes (Huang et al., 2023). Xi’an, situated in central-west China, has witnessed an increase imported cases in the past few years, raising concerns of local dengue transmission. It is crucial to investigate the population density of Aedes albopictus and its insecticides resistance to enhance early warning of dengue fever.

Methods

Eight sampling sites in eight counties (YT, BL, WY, CH, YL, LN, LT, ZZ) of Xi’an city were surveyed by larval dipping and human-baited double net trap biweekly from June 2021 to September 2022. The Breteau Index (BI, number of positive containers per 100 houses) and Container Index (CI, the percentage of containers containing larvae or pupae) were used to assess larval density, and the human-baited double net trap (HDN, the number of Ae. albopictus females collected per person per hour) to indicate human bating rate (HBR). Meanwhile, the association between the meteorological factors and mosquito density was analyzed. The Ae. albopictus adult insecticides resistance was evaluated by the World Health Organization (WHO) standard resistance bioassay. Adult females were exposed to insecticide-impregnated paper for 1 h, then transferred to the recovery tube, and mortality rate was calculated after 24 h. According to the Implementation Plan for National Vector Surveillance (2016), resistance status was classified into three levels based on mortality: <80% mortality as resistant, between 80% and 98% mortality as possibly resistant, and >98% mortality as sensitive.

Results

From June 2021 to September 2022, a total of 1,065 houses were surveyed for water holding containers, and 99 of 430 water holding containers were checked to be positive for Ae. albopictus larvae and pupae. A total of 1,048 Ae. albopictus females were collected. The average BI, CI and HBR were 10.39, 21.41, and 11.20 female/man/hour in 2021 and 8.86, 20.86, and 11.63 f/m/h in 2022, respectively. The findings showed that the BI exceeded 5 in most months and reached above 20 in specific months. The CI varied in different months and monitoring sites, with the highest CI in August 2021 and July 2022. The discarded tires had the highest positivity rate, with up to 40.32% testing positive for Ae. albopictus larvae. The monthly average temperature showed a positive correlation with CI (r = 0.77), and the monthly BI was positively associated with CI (r = 0.93). The BI, CI, and HBR were significantly higher in the rainy season than other seasons. The bioassay results showed that the mortality rate of Ae. albopictus at the YT monitoring site was 76.92%, indicating resistance to deltamethrin. The mortality rate of Ae. Albopictus at BL, WY, CH, YL, LN, LT, and ZZ sampling sites were varying from 81.25%∼100%, suggesting possibly resistant or still sensitive to beta-cypermethrin, alpha-cypermethrin, malathion, chlorpyrifos, and propoxur.

Introduction

With global warming and increased urbanization, mosquito-borne diseases such as dengue have become an important global concern, posing a significant strain on national healthcare systems and economic development (Franklinos et al., 2019). Dengue is the deadliest mosquito-borne disease after malaria, and nearly half of the global population are at risk of dengue, with an estimated 100 to 400 million people infected with dengue each year (WHO, 2024). It was epidemic in more than 100 countries, including Southeast Asia, Central and South America, Africa, the Western Pacific, and the Eastern Mediterranean (Wilder-Smith, Murray & Quam, 2013). Since the 1970s, several dengue outbreaks have occurred in the southern provinces of China, including Guangdong, Fujian, Yunnan and Guangxi. These outbreaks were mainly local outbreaks caused by imported cases with seasonal and endemic characteristics (Lun et al., 2022). However, in recent years, some traditional northern cities, such as Puyang and Jining (Huang et al., 2023), have experienced indigenous dengue outbreak. With the accelerated urbanization, increased international trade and frequent travel in China, it is expected that new cases of local dengue fever will be emerging in other cities.

Aedes albopictus (Skuse, 1894), also known as the Asian tiger mosquito, transmits Dengue, Chikungunya, Zika and Yellow fever and is one of the world’s most invasive species due to its ability to adapt to new environments (Bonizzoni et al., 2013; Houé et al., 2019). Native to Southeast Asia, Ae. albopictus has invaded Africa, Europe, and the Americas over the past 30 years (Ngoagouni et al., 2015; Garcia-Rejon et al., 2021; Peach & Matthews, 2022). In China, it has been found not only in warm southern regions, such as Hainan Islands and Guangdong Province, but also in colder regions, such as Tibet Autonomous Region and Liaoning Province (Liu et al., 2020).

In the absence of an effective vaccine and specific treatment for dengue fever, mosquito vector control remains the main measure to prevent outbreaks of mosquito-borne diseases, and insecticides are an important means to reduce mosquito density (Paz-Bailey et al., 2024). Long-term irrational use of chemical insecticides, on the one hand, pollutes the environment, and at the same time causes resistance, which also leads to the decreasing effectiveness of treatment (Hou et al., 2020). With more and more dengue epidemics occurring in inland cities in the north, Xi’an, as a world-famous tourist city with frequent foreign exchanges, coupled with the widespread distribution of the vector Ae. albopictus mosquito, has prepared to cope with the occurrence of dengue fever. To grasp the characteristics of the spatial and temporal distribution of dengue vectors in the region, as well as the development of their resistance status to common insecticides, we conducted this surveillance, which will provide a scientific basis for dengue risk assessment and early warning.

Materials and Methods

Study area

Xi’an City is in central-west China, which is located between 107°40′ and 109°49′E in longitude, between 33°42′ and 34°45′N in latitude (Fig. 1). The altitude spans a wide range from 345 to 3,626 m. The Xi’an city is bordered by the Wei River to the north and the Qinling Mountains to the south and is rich in flora and fauna. With a temperate continental climate, the average annual temperature ranges from 11 to 18 °C, and the annual precipitation is from 550 to 950 mm. Rainfall is plentiful but very unevenly distributed, with 60–70% of annual precipitation concentrated in the summer months (Hou, Wu & Xie, 2020).

Figure 1 Study area: map of sampling sites in Xi’an, China.

(A) The location of the study area in China. (B) The sampling sites in Xi’an. The figure was produced in ArcGIS 10.8.1 (ESRI, Redlands, CA, USA) using shape files representing Xi’an City, which were obtained from Map Technical Review Centre, Ministry of Natural Resources of China, and the base map is unmodified. Review No. GS (2020) 4619.

Mosquito ecological surveillance

Eight monitoring sites (YT, BL, WY, CH, YL, LN, LT, ZZ) were selected for mosquito surveillance (Data S1). The type of environment in each monitoring site includes residential, parks, and the recycling stations or construction sites.

The Breteau Index (BI) and Container Index (CI) were selected to assess larval density. All the water holding containers were checked in and out the houses. The larvae in the positive containers were sucked through droppers and collected into a beaker for species identification, then transported to the laboratory and fed with dechlorinated tap water and mouse chow until adult, which were prepared for resistance test. Aedes larvae were maintained under laboratory conditions at a temperature of 27 ± 2 °C, relative humidity of 75 ± 10% and a cycle of 14 h of light and 10 h of darkness.

Adult mosquito surveillance was conducted by the human-baited double net trap. Human biting rate (HBR) was calculated by Ae. albopictus females collected per person per hour. Each environment to choose a sheltered location to utilize a net trap, in the vector Ae. albopictus activity peak hours (15:00–18:00). The human bait stood in the internal nets with bared arms and legs. The collector used a mosquito respirator to collect Ae. albopictus adults landing on the net for 30 min. The captured adults were stored in a refrigerator and freeze to death. Then sex, species were identified, and number was counted. At the same time, the sampling site, date, temperature, humidity and wind speed should be recorded in a sheet. The monthly average temperature and rainfall data were obtained from the National Earth System Science Data Centre (http://www.geodata.cn) (Peng et al., 2019).

Adult resistance bioassay

Adult resistance bioassay were performed by diagnostic doses according to the adult mosquito contact bucket method recommended by the World Health Organization (WHO, 2003). Test adults females were from the larval surveillance, that were fed in rearing room until adults. Twenty female mosquitoes of 3∼4 days old after eclosion were placed in the exposure tube with a sheet of insecticide-impregnated paper for 1 h and then transferred to the recovery tube. The mosquitoes were examined for death after 24 h of feeding with 10% glucose water, and the mortality was calculated. The test was repeated three times and the control group mosquitoes were treated with a sheet of oil-impregnated control paper by the same procedure. Six insecticides from three categories (pyrethroids, organophosphates, and carbamates) were tested in the bioassay, including deltamethrin (0.03%), beta-cypermethrin (0.1%), alpha-cypermethrin (0.03%), malathion (0.5%), chlorpyrifos (2%), and propoxur (0.05%).

The temperature, humidity and light condition in the test room were the same as in the rearing room. Both test and control groups consisted of non-blood-sucking F2 generation female mosquitoes. Mosquitoes that were unable to fly under mechanical stimuli were considered dead. According to the Implementation Plan for National Vector Surveillance (2016), resistance status was classified into three categories based on mortality: <80% mortality as resistant, between 80% and 98% mortality as possibly resistant, and >98% mortality as sensitive (Zhao et al., 2022). Abbott’s formula was applied for corrected mortality if the control mortality rate was ≥5% and <20%. If the control mortality rate is ≥20%, the bioassay should be repeated. The Abbott formula is: Corrected mortality%=% mortality with treated paper−% mortality with control1−% mortality with control×100.

Data analysis

Data was calculated by Excel2021 and analyzed by IBM SPSS Statistics V24.0 (SPSS ver 24.0, Armonk, NY, USA). Levene’s analysis was used to test the heterogeneity of variance for each group; Independent samples t-test was used to test the BI, CI and HBR between 2021 and 2022; chi-square test was used to analyze the differences of different types of containers; one-way ANOVA or Welch’s ANOVA was performed for the BI, CI and HBR during different months and different monitoring sites; two-by-two tests were performed by SNK-q or Dunnett’s T3; Pearson correlation analysis was used to test the interaction between different insecticides. All the test levels were set to α = 0.05.

Results

Breteau index

The results of the spatial and temporal distribution of BI were shown in the heatmap (Fig. 2A; Data S2). A total of 1,065 households were surveyed during the surveillance period, and 103 Aedes positive containers (23.95%) were found in 430 water holding containers. The average BI for 2021 and 2022 were 10.39 and 8.86, respectively. There was no difference in BI between 2021 and 2022 by the independent samples t-test (t = 0.962, P = 0.341). Ae. albopictus larvae densities showed seasonal fluctuations.

Figure 2 The spatio-temporal characteristic and monthly variation of Aedes albopictus (larva and adult indices) between 2021 and 2022.

(A) Spatio-temporal variation of BI between 2021 and 2022. (B) spatial-temporal variation of CI between 2021 and 2022. (C) Spatial-temporal variation of Aedes albopictus females collected per man per hour by the HDN method (unit: f/m/h) between 2021 and 2022. (D) The monthly average BI between 2021 and 2022. (E) The monthly average CI between 2021 and 2022. (F) The monthly average adult females collected per month between 2021 and 2022.

Levene’s test showed that there is homogeneity of variance across different months (all P > 0.01). After one-way ANOVA, the results of BI in different months were statistically different (2021: F = 7.870, P < 0.05; 2022: F = 6.260, P < 0.05). The highest monthly BI in 2021 (18.68) was in August, and the highest monthly BI in 2022 (13.33) was in July (Fig. 2D). The highest BI value (33.33) during the survey period occurred in August 2021 at the WY monitoring site.

Container index

The results of the spatial and temporal distribution of CI were shown in the heatmap (Fig. 2B; Data S3). A total of 430 water holding containers were investigated during the two years, and the main types included bonsai/aquatic plants, basin, idle containers, canal rockery, bamboo or tree holes, discarded tires and green belts (Table 1). Chi-square analysis showed a difference in the positive rate of different containers (χ2 = 15.50, P < 0.05). The positivity rate of discarded tires was 40.32%, which was higher than other container types. The mean CI was 21.41 and 20.86 in 2021 and 2022, respectively. the CI showed no difference between the two years by independent samples t-test (t = 0.165, P = 0.869). By one-way ANOVA, the CI was statistically different between months (2021: F = 3.569, P < 0.05; 2022: F = 3.636, P < 0.05). The highest monthly CI was found in August 2021, and July 2022 (Fig. 2E). The highest CI during the survey period occurred at the BL monitoring site in July 2021, at 62.50 (Fig. 2B). Welch’s ANOVA showed that there were differences in containers among the different monitoring sites (F = 2.893, P < 0.05). Dunnett-T3 test showed that the CI at the WY monitoring site was higher than the LN monitoring site.

Table 1 The presence of Aedes larvae and pupae in different types of water holding containers.

Types of containers	Positive	Negative	
	Number	Rate (%)	Number	Rate (%)	
Bonsai	17	20.00	68	80.00	
Basin	18	22.78	61	77.22	
Idle container	23	28.05	59	71.95	
Canal rockery	10	16.67	30	83.33	
Bamboo/tree holes	7	21.21	26	78.79	
Discarded tired*	25	33.87	37	66.13	
Green belt	3	10.34	26	89.66	
Notes.

* P < 0.05.

Human biting rate

The human biting rate (HBR) was conducted by the human-baited double net trap method, and the spatial and temporal distribution of human biting rate is shown in the heatmap (Fig. 2C; Data S4). A total of 184 net trap were deployed in the two years, and 1,048 female Ae. albopictus mosquitoes were captured, all of which were identified as Ae. albopictus. The average human biting rate during the monitoring period was 11.39 (f/m/h). Among them, the human biting rate in 2021 and 2022 were 11.20  ± 4.44 (f/m/h) and 11.63 ± 4.27 (f/m/h), respectively. There was no difference in human biting rate between the two years by one-sample t-test. There were differences in human biting rate between months by ANOVA, (2021: F = 10.856, P < 0.05; 2022: F = 5.119, P < 0.05). The monthly average human biting rate was highest in July 2021, 16.29 ± 3.37 (f/m/h); and in August 2022, 15.09 ± 3.38 (f/m/h) (Fig. 2F). ANOVA test shown that human biting rate differed among monitoring sites (F = 5.473, P < 0.05). The human biting rate at the WY and BL monitoring sites were higher than that of LT and LN by SNK-q test.

Climate impacts

The study area entered the rainy season in July with a significant increase in temperature and rainfall. After Pearson correlation analysis, there was a correlation between meteorological factors and mosquito density indicators (Fig. 3). The correlation coefficient between temperature and CI was 0.77 (Fig. 3B, P < 0.05), and the correlation coefficient between BI and CI was 0.93 (Fig. 3C, P < 0.05).

Figure 3 Pearson correlation analyses between Ae. albopictus density and meteorological factors.

(A) Correlation analysis between BI and temperature. (B) Correlation analysis between CI and temperature. (C) Correlation analysis between BI and CI. (D) Correlation analysis between HBR and temperature.

There was a difference in the density index of Ae. albopictus between the rainy season (July and August) and the dry season (June and September) (Table 2). The annual BI was 12.90 in the rainy season compared to 6.34 in the dry season. The BI was higher in the rainy season than in the dry season by the independent samples t-test (t =  − 4.802, P < 0.05); and the mean CI was 27.58 in the rainy season compared to 17.09 in the dry season. The CI was also higher in the rainy season than in the dry season by the independent samples t-test (F =  − 3.971, P < 0.05). Significantly more adult mosquitoes were also collected in the rainy season than in the dry season, accounting for 60.60% (635/1,048) of the total captured mosquitoes. The human biting rate in the rainy season was also higher than in the dry season by independent samples t-test (F = 5.841, P < 0.05).

Insecticide resistance

The insecticide resistance status in different sampling sites were shown in Fig. 4. The resistance results showed that adult mosquitoes exhibited varying degrees of decreased susceptibility to three categories of insecticides (pyrethroids, organophosphates and carbamates). The distribution of insecticide resistance to pyrethroids was significantly higher than that of carbamates and organophosphates. For alpha-cypermethrin, beta-cypermethrin and deltamethrin in the pyrethroid group, all monitoring sites except BL and CH were resistant or possibly resistant to pyrethroids. Three monitoring sites showed probable resistance to propoxur, with mortality ranging from 87.78% to 95.56%; five sites showed resistance and probable resistance to deltamethrin, with Ae. albopictus at the YT site showing resistance to deltamethrin (76.92%); four sites showed probable beta-cypermethrin resistance; 3 sites with probable resistance to malathion, with mortality ranging from 91.46% to 96.15%; 5 sites with probable resistance to alpha-cypermethrin; and 3 sites with probable resistance to chlorpyrifos, with mortality ranging from 93.48% to 96.00%.

Table 2 Aedes density indictors between rainy season and dry season in 2021 and 2022.

Year	BI	CI	HBR	
	Rainy	Dry	Rainy	Dry	Rainy	Dry	
2021	14.65	6.16	30.08	18.48	13.91	14.00	
2022	11.32	6.37	27.52	16.67	8.46	9.33	

Figure 4 Resistance status of Aedes albopictus to six insecticides in Xi’an.

(A) The resistance status of Ae. albopictus to propoxur. (B) The resistance status of Ae. albopictus to deltamethrin. (C) The resistance status of Ae. albopictus to beta-cypermethrin. (D) The resistance status of Ae. albopictus to malathion. (E) The resistance status of Ae. albopictus to alpha-cypermethrin. (F) The resistance status of Ae. albopictus to chlorpyrifos. Created in ArcGIS 10.8.1 (ESRI, Redlands, CA, USA) using shape files representing Xi’an City, which were obtained from Map Technical Review Centre, Ministry of Natural Resources of China. The base map is unmodified. Review No. GS (2020) 4619.

The bioassay results were shown in Data S5. Mortality correlation between different insecticides were carried out by Spearman analyses (Fig. 5). The results showed that the mortality of deltamethrin was positively correlated with that of alpha-cypermethrin with correlation coefficient r = 0.87 (P < 0.05). The mortality of propoxur was negatively correlated with that of alpha-cypermethrin (r =  − 0.720, P < 0.05). Overall, Ae. albopictus adult has developed more severe resistance to pyrethroids, but results in most monitoring sites remained at a low resistance or showed to be still sensitive to carbamate and organophosphate insecticides.

Figure 5 The Spearman correlation analysis between Ae. albopictus mortality to different insecticides.

Discussion

In recent years more and more inland Chinese cities, such as Xi’an and Zhengzhou, have imported dengue cases, and some traditional northern cities have experienced indigenous dengue outbreaks. The dengue vector Ae. albopictus is widely distributed in Xi’an, and it is particularly important to monitor the vector Ae. albopictus. This study is the most comprehensive study on Ae. albopictus in Xi’an at present, analyzing the larval indices, adult density and resistance status of Ae. albopictus during 2021–2022.

The BI is an important indicator for assessing the density of Ae. albopictus and the risk of community transmission during dengue epidemics: BI <5 (low infection rate, low risk of dengue transmission); BI ≥ 5 (risk of transmission); BI ≥ 10 (risk of outbreak); and BI ≥ 20 (risk of regional epidemics) (WHO, 2003). In both 2021 and 2022, the average BI in June was greater than 5, which indicated a risk of dengue transmission according to the criteria. The BI rose quickly during July as the temperature and rainfall increased; in August, it even approached 20, indicating a possible dengue outbreak in the area. During the monitoring period, local areas, such as BL (Jul2021, Aug2021) and YL (Aug2021), had a BI of over 20, which had an outbreak or even a regional epidemic risk in the event of imported dengue cases.

The highest average BI was recorded in monitoring site WY (15.24), probably because this area is in rapidly developing and there are many construction sites with various idle container on the ground. The average BI was also higher in monitoring site BL (14.13), which may because the site is in an old urban district, whose public infrastructures are generally outdated, with piles of idle objects, and a lack of property management. On the other hand, there are more elderly people in these residential neighborhoods, whose awareness of hygiene is insufficient, and who have the habit of collecting rainwater for watering flowers and so on. All these create convenient conditions for mosquito breeding.

Mosquito density varied from month to month due to meteorological factors such as temperature, humidity and rainfall (Liu et al., 2023). The findings showed a positive correlation between temperature and BI and CI (Fig. 3), which is similar to the findings of De Resende et al. (2013). They noted that temperature was positively correlated with adult and larval collection. However, in the present study, no significant correlation was found between temperature and adult mosquito density. This could be attributed to the susceptibility of the human-baited double net trap method to a range of factors, including wind speed, the physical condition of the human bait, and the location of the monitoring site. Whereas the effect of rainfall on adult mosquito density remains uncertain, Camargo et al. (2021) pointed out a positive correlation between adult mosquito density and precipitation (Wang et al., 2020). This correlation has not been observed in this study.

Waste recycling stations/construction sites are prone to mosquito breeding due to the presence of many water holding containers that are left uncleaned for long periods of time. Tire stagnant water is not easily drained and is an ideal breeding ground for Ae. albopictus (Gratz, 2004). Even if the water evaporates from these tires, the eggs left in the sludge at the bottom do not die and resume breeding as soon as the amount of water and temperature are suitable (Susong et al., 2022). With the export of discarded tires to other countries and regions, the global spread of Ae. albopictus (González et al., 2020; Mohammadi et al., 2022). The present survey showed a positive rate of 40.32% for discarded tires, which is higher than other types of containers (Table 1). It is similar to the findings of Song et al. (2022) in Zhejiang Province, China, but lower than the results (>80%) in African countries such as Tanzania and Ghana (Joannides et al., 2021; Ngingo et al., 2022). These discarded tires are mainly found in densely populated places such as parks, marinas, automobile repair workshops, outdoor playgrounds, and bed and homestay, which need extra attention. Various types of idle water containers, including plastic and glass containers, are also important places for Aedes breeding, and the mean CI of various types of plastic containers in this survey was 23.95. Since these containers are mostly distributed in some unattended places and corners, they are very likely to cause excessive mosquito density in the area (Ren et al., 2018).

Mosquito-borne diseases such as dengue fever are transmitted by Aedes adult female through bloodsucking, therefore Aedes adult female surveillance is more suitable for risk assessment than larval indices (Yang et al., 2021). It has been suggested that the human-baited double net trap may underestimate the true density of Ae. albopictus due to the structure of the double-layered net. The results of this surveillance showed that the highest density of adult mosquitoes was found in July, which reached 14.68. There was a difference in the density of adult mosquitoes between different monitoring sites (F = 5.47, P < 0.01), with the highest of 16.33 in the WY monitoring site.

Pyrethroids, organophosphates, and carbamates are the most used insecticides in pest management, with pyrethroids being the notably prevalent in domestic mosquito control (López Dávila et al., 2020).With the widespread use of insecticides, the problem of insecticide resistance has become more and more prominent. A study showed that the Aedes in southern Benin has developed either resistant or suspected resistant to all tested pyrethroids (Konkon et al., 2023). Li et al. (2021) reported that Ae. albopictus in Hainan Province was extensively resistant to a variety of insecticides. Wei et al. found that the resistance of Ae. albopictus to pyrethroid insecticides increased significantly after a short period of extensive use of insecticides (Ling-ya et al., 2019). The results of this resistance bioassay showed that Ae. albopictus at the YT monitoring site, had developed resistance to deltamethrin (76.92%). This aligns with the findings from investigations conducted in provinces such as Anhui and Hainan in China, indicating that Ae. albopictus has developed a widespread resistance to pyrethroids (Hou et al., 2023; Wu et al., 2024). Meanwhile, the Ae. albopictus at these sites were still to be sensitive or possibly resistant to organophosphates, and carbamates, with mortality rate varying from 81.25% to 100%. The results were different from West Bengal and Latin America, in which the Ae. albopictus population have developed severe resistance to organophosphates and carbamates (Asgarian et al., 2023). When adult mosquitoes become resistant to an insecticide, its likely to be resistant to insecticides with the similar toxicity mechanisms, which is known as cross-resistance (Moyes et al., 2021). The results of the present resistance experiment showed a positive correlation (r = 0.870) between the lethality of deltamethrin and alpha-cypermethrin, suggesting the existence of cross-resistance between them. The development of insecticide resistance can be attributed to metabolic resistance and target-site mutations (Machani et al., 2020). Further studies are needed to further understand changes at molecular and microstructural level.

Ae. albopictus is a semi-domesticated mosquito species that mainly breeds in a variety of small water holding containers, and the removal of breeding sites should be the primary measurement (Zhang et al., 2023). Reducing the Aedes breeding inhabitants by turning over pots and pouring irrigation or breeding ornamental fish that prey on mosquito larvae can effectively reduce the density of larvae. Biological control methods such as Wolbachia, as a new vector control strategy, have also been successful applied in some areas (Fox et al., 2024). We recommend local governments and health organizations to adopt an integrated management approach combining environmental treatment and chemical control before the peak of Ae. albopictus larvae in July, and thus reduce the risk of mosquito-borne diseases.

Conclusions

We analyzed the seasonal fluctuation of Ae. albopictus and its resistance to insecticides, which was critical for dengue fever early warning. Based on the results and discussion, some main conclusions can be drawn as follows:

(1) The BI consistently exceeded the threshold value during the surveillance period, indicating a persistent risk of dengue transmission. Especially during July and August, the surge in BI indicates a potential outbreak risk in the high-risk areas.

(2) The Ae. albopictus density demonstrated seasonal fluctuations, but no significant differences were observed between the years.

(3) The CI for discarded tires was significantly higher than that of other water holding containers, indicating that discarded tires were major breeding sites for Ae. albopictus and required additional attention.

(4) There were differences in HBR between the different monitoring sites, with the highest HBR at the WY site, indicating a higher density of adult mosquitoes in the area.

(5) The temperature was positively associated with CI, highlighting how meteorological factors greatly influence Ae. albopictus population, especially during the rainy season.

(6) The Ae. albopictus adults has developed resistance to deltamethrin at the YT site, reflecting a growing trend of insecticide resistance.

Therefore, mosquito vector surveillance and integrated mosquito control should be continuously strengthened, especially in July before the peak of larva reproduction, to decrease the possibility of future dengue transmission.

Supplemental Information

Data S1 The coordinate information of the eight sampling sites in Xi’an, China

Data S2 The BI value from Jun. 2021 to Sep. 2022

Data S3 The CI value from Jun. 2021 to Sep. 2022

Data S4 Number of Ae. albopictus females collected per man per hour by the HDN method

Data S5 The results of Ae. albopictus females resistance to six insecticides

Additional Information and Declarations

Competing Interests

Author Contributions

Data Availability

The authors declare there are no competing interests.

Xiaogang Lei conceived and designed the experiments, prepared figures and/or tables, authored or reviewed drafts of the article, and approved the final draft.

Songtao Pang conceived and designed the experiments, authored or reviewed drafts of the article, and approved the final draft.

Qipeng Zhang analyzed the data, prepared figures and/or tables, and approved the final draft.

Kun Xu performed the experiments, authored or reviewed drafts of the article, and approved the final draft.

Wei Xue performed the experiments, analyzed the data, authored or reviewed drafts of the article, and approved the final draft.

Mingxu Wu performed the experiments, analyzed the data, prepared figures and/or tables, and approved the final draft.

Xiangdong Li performed the experiments, prepared figures and/or tables, and approved the final draft.

Liangdong Jin performed the experiments, authored or reviewed drafts of the article, and approved the final draft.

Guangshuai Li performed the experiments, authored or reviewed drafts of the article, and approved the final draft.

Baozhong Chen conceived and designed the experiments, authored or reviewed drafts of the article, and approved the final draft.

The following information was supplied regarding data availability:

The raw data are available in the Supplementary Files.

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
