# Peer review of "Ecological features and insecticide resistance of Aedes albopictus in Xi’an, a high-risk dengue transmission area in China"

_PeerJ, doi:10.7717/peerj.18246_

## Round 0.1 · original submission · Minor Revisions

Please pay attention to the specific comments of the reviewers. The article should attract the attention of specialists in this topic all over the world. It is necessary to cite sources on this insect species more widely within its entire range. Overall, the article is quite informative and I believe it will find its readers.

·

Basic reporting

Aedes albopictus are carriers of serious diseases, including Dengue fever, as well as West Nile, Zika and other viruses. Due to climate change, this invasive species is already common in Mediterranean countries. It is expected that in the coming decades they will spread to northern Europe. Methods for combating this type of mosquito are being developed more intensively in Asian countries, where they are endemic. Therefore, the topic of the peer-reviewed article is very relevant. The research questions and hypotheses in the manuscript are clearly defined. The conclusions are supported by statistical analysis. The manuscript is formatted according to the requirements. However, there are some minor flaws and technical comments.

I recommend amending the title of the article, replacing the term "bionomics" with a more conservative one - "ecological features". Since a significant part of the work is devoted to the study of the population density of larvae and adults of Aedes albopictus in Xi'an and laboratory observations.

At the first mention of the Latin name of the insect in the text of the manuscript, it is necessary to indicate the author's surname and the year of description of the species.

In the "Discussion" section, the literary information on the sensitivity of Aedes albopictus to various classes of insecticides is insufficiently analyzed. I recommend adding.

Experimental design

On the basis of what methodology was the status of mosquito resistance to insecticides classified? There is no literature reference. Is this the authors' own development? Up to 80% mortality in resistant individuals is a high mortality rate. On what basis were such criteria for the status of resistance adopted? If the authors of the manuscript were guided by a standard methodology, it is necessary to add a literature reference in the text.

In the "Materials and Methods" section, it is necessary to indicate which insecticides were used in the research.

In the text of the manuscript, some sentences are missing spaces (94, 218, 229, etc.).

Validity of the findings

Why were pyrethroid, carbamate and organophosphorus insecticides chosen for the research? No justification. I recommend adding.

I recommend expanding the “Conclusion” section and making it more informative.

Additional comments

No comments.

·

Basic reporting

The manuscript is devoted to mosquito vector surveillance of dengue transmission and integrated mosquito control. This problem is relevant for China, because there was witnessed an increase of imported cases of the disease.
English language used throughout is clear and professional.
The abstract reflects the background, methods and content of the conducted research. Your abstract needs more details. I suggest you to increase the abstract to 500 words (3000 characters). Тhe references are adequate and appropriate. 66 % of the referenced literature is new, published in the last 5 years. The structure of the manuscript conforms to PeerJ standards. Figures successfully illustrate the text of the research. They are relevant, high quality and well described. This manuscript has raw data supplied (in the folder “supplemental”).

Experimental design

The research is original. Research question well defined, relevant and meaningful. It expands understanding of mosquito biology, ecology and effective methods of Ae. albopictus population control.
The investigation was performed in accordance with a high technical and ethical standard. Methods were described with sufficient detail. Modern methods of ecological analysis were used in this work: the Breteau Index (BI) and Container Index (CI) were used to assess larval density; the human-baited double net trap (HDN) to indicate human bating rate (HBR). The association between the meteorological factors and mosquito density was analyzed. The resistance of adults of Ae. albopictus to insecticides was studied.

Validity of the findings

Conclusions are well stated, linked to original research question and limited to supporting results. All underlying data have been provided; they are reliable, statistically sound, and controlled.

Additional comments

I commend the authors for their extensive data set, compiled over 2 years of detailed fieldwork.
Tips for improving your manuscript.
1. I suggest you to increase the abstract to 500 words (3000 characters).
2. Line 44. Replace Aedes albopictus with Aedes albopictus (Skuse, 1894). The first mention of any organism must include the full scientific name with the author and the year of publication.
3. Lines 283-284. The authors write: “We recommend local governments and health organizations to adopt an integrated management approach combining environmental treatment, biological control and chemical control before …”. The mention of biological control is inappropriate, because the authors do not analyze it in the manuscript.
4. I propose to increase the conclusions and divide them into 6 parts: 1. conclusion according to the Breteau index, 2. results according to Levene's test, 3. conclusion according to the container index. 4. the results of the HBR analysis, 5. the effect of climate on the mosquito population, 6. resistance to insecticides.
Therefore, the work is relevant, meaningful and can be recommended for printing with minor corrections.

---

## Round 0.2 · accepted · Accept

Dear colleagues, I thank you for your good work on your manuscript and recommend it for publication.

·

Basic reporting

The authors of the manuscript took the reviewers' comments very seriously and made all the corrections to the text. The manuscript has acquired a finished look. The authors have conducted a large and interesting study. The article is written at a high scientific level and is devoted to a relevant topic. The conclusions are confirmed by statistical analysis. Mathematical data processing was carried out using the most modern software. The work is formatted according to the requirements. The manuscript is quite original, and after making several additions, it can be recommended for publication.


At the first mention of the Latin name of the insect in the abstract, it is also necessary to indicate the author's surname and the year of the species description.

Experimental design

No comments.

Validity of the findings

I recommend adding the phrase "The studied populations of Aedes albopictus mosquitoes are generally resistant to pyrethroid insecticides and moderately sensitive to carbamates and organophosphates" (from the "Results" section).

·

Basic reporting

no comment

Experimental design

no comment

Validity of the findings

no comment

Additional comments

I thank the authors. They took into account all my comments. I think that the manusсript can be recommended for publication.